# A conformation-locking inhibitor of SLC15A4 with TASL proteostatic anti-inflammatory activity

Andras Boeszoermenyi [1], Léa Bernaleau[2,7], Xudong Chen[3,7], Felix Kartnig[1,4], Min Xie[3], Haobo Zhang [2], Sensen Zhang[3], Maeva Delacrétaz [2], Anna Koren[1], Ann-Katrin Hopp[1], Vojtech Dvorak[1], Stefan Kubicek [1], Daniel Aletaha [4], Maojun Yang[3,5], Manuele Rebsamen [2,8] ✉, Leonhard X. Heinz [4,8] ✉ & Giulio Superti-Furga [1,6,8] ✉

Dysregulation of pathogen-recognition pathways of the innate immune system is associated with multiple autoimmune disorders. Due to the intricacies of the molecular network involved, the identification of pathway- and disease-specific therapeutics has been challenging. Using a phenotypic assay monitoring the degradation of the immune adapter TASL, we identify feeblin, a chemical entity which inhibits the nucleic acid-sensing TLR7/8 pathway activating IRF5 by disrupting the SLC15A4-TASL adapter module. A high-resolution cryo-EM structure of feeblin with SLC15A4 reveals that the inhibitor binds a lysosomal outward-open conformation incompatible with TASL binding on the cytoplasmic side, leading to degradation of TASL. This mechanism of action exploits a conformational switch and converts a target-binding event into proteostatic regulation of the effector protein TASL, interrupting the TLR7/8-IRF5 signaling pathway and preventing downstream proinflammatory responses. Considering that all components involved have been genetically associated with systemic lupus erythematosus and that feeblin blocks responses in disease-relevant human immune cells from patients, the study represents a proof-of-concept for the development of therapeutics against this disease.

Recognition of pathogen-derived nucleic acids by pattern recognition receptors is essential to mount protective innate immune responses[1–5]. Despite tight regulatory mechanisms, aberrant activation of these pathways by endogenous ligands or mutations in key regulatory components leads to excessive responses that are causatively linked to

several autoimmune conditions[6,7]. In particular, sensing of nucleic acids by endolysosomal TLRs is thought to be critically involved in the etiology of systemic lupus erythematosus (SLE) and related diseases[8–11]. Both human genetic and mouse studies have unequivocally identified the lysosomal solute carrier SLC15A4 and transcription factor IRF5 as

[1]CeMM Research Center for Molecular Medicine of the Austrian Academy of Sciences, Vienna, Austria. [2]Department of Immunobiology, University of Lausanne, Epalinges, Switzerland. [3]Ministry of Education Key Laboratory of Protein Science, Tsinghua-Peking Center for Life Sciences, Beijing Advanced Innovation Center for Structural Biology, School of Life Sciences, Tsinghua University, Beijing, China. [4]Division of Rheumatology, Department of Internal Medicine III, Medical University of Vienna, Vienna, Austria. [5]Cryo-EM Facility Center, Southern University of Science & Technology, Shenzhen, Guangdong, China. [6]Center for Physiology and Pharmacology, Medical University of Vienna, Vienna, Austria. [7]These authors contributed equally: Léa Bernaleau, Xudong Chen. [8]These authors jointly supervised this work: Giulio Superti-Furga, Leonhard X Heinz, Manuele Rebsamen. ✉e-mail: manuele.rebsamen@unil.ch; leonhard.heinz@meduniwien.ac.at; gsuperti@cemm.oeaw.ac.at

components essential for mediating disease development downstream of TLRs[12–22]. Investigating the mechanistic involvement of SLC15A4, we recently discovered the protein TASL, encoded by an SLE-associated gene previously known as *CXorf21*, as an interactor essential for IRF5 activation[13,23,24]. Through a C-terminal pLxIS motif, TASL acts as signaling adapter mediating recruitment of IRF5, in analogy to the key immune adapters MAVS, STING and TRIF for IRF3[23,25]. Thus, TASL represents the fourth, central element in a pathway in which each component is associated with SLE, providing an unusually strong case and rationale for therapeutic intervention. As our previous data had shown that interfering with SLC15A4-TASL complex formation abolished TLR-induced IRF5 activation, this suggested a strategy to target the pathway with high specificity. Conveniently, TASL is regulated by proteostatic interaction with SLC15A4, so that interfering with complex formation alone leads to efficient degradation of TASL protein (Fig. 1a)[23].

In this study we investigated the possibility of exploiting this property for chemical targeting and report the identification of a chemical compound that efficiently promotes TASL degradation and inhibition of downstream responses. The unusual mechanism of action revealed by the cryo-EM structure involves a conformation-specific interaction with the lysosomal outward-open conformation of SLC15A4, which is incompatible with TASL binding, leading to the proteolytic removal of this essential signaling component.

## Results

### Molecular definition of the SLC15A4-TASL binding interface

To better explore the molecular basis of the binding of TASL to SLC15A4, we modeled the SLC15A4/TASL complex with AlphaFold2 (Fig. S1a)[26,27]. While much of the TASL structure was poorly defined, in the highest-ranked model, an α-helix at its N-terminus (formed by residues 1-23) intruded deep into the peptide binding cavity of SLC15A4 in an inward-open conformation (Fig. 1b). This was consistent with our previous interaction studies, which showed that mutation of the polar residues in the first eight amino acids of TASL abolished complex formation[23]. Moreover, in the AlphaFold model, glutamate 465 (E465) in SLC15A4, required for TASL binding[23], was in close proximity to the TASL N-terminus. Of note, in the related peptide transporters SLC15A1/2, the glutamate corresponding to E465 mediates interactions with the N-termini of the cargo peptides[28]. Mutation of E465 has been previously used to impair SLC15A4 transport and to support the notion that SLC15A4 transport activity is required for TLR7-9 responses[15]. To assess the overall binding mode suggested by this model, we selected a series of SLC15A4 residues predicted to contribute to TASL binding, as well as a few adjacent amino acids as controls, for a focused alanine scanning mutagenesis analysis (Fig. 1c). Co-immunoprecipitation experiments in SLC15A4-deficient THP1 cells reconstituted with these variants strongly support the validity of the model with residues observed contacting TASL impairing binding, while adjacent substitutions having no or minor effects (Fig. 1c). Lastly, we assessed whether the N-terminal portion of TASL, that in the AlphaFold prediction intrudes into the SLC15A4 channel, was sufficient to mediate binding. SLC15A4 co-immunoprecipitated a peptide representing the first 13 amino acids of the protein, demonstrating that the N-terminus of TASL is indeed sufficient for the interaction (Fig. 1d). In contrast, SLC15A4 failed to recruit TASL (1-12), possibly because of a steric hindrance mediated by the C-terminal GFP, which we hypothesize to prevent the shorter peptides of inserting sufficiently deep into the SLC15A4 channel. Collectively, these data are compatible with an unprecedented mode of binding between a protein ligand and a solute carrier. This model suggests that TASL would be able to bind the cytoplasmic inward-open, but not the lysosomal outward-open SLC15A4 conformation. The extent and nature of the observed mode-of-interaction, together with the outcome of mutagenesis and interaction studies, warranted the validity of the model and revealed an opportunity for chemical targeting.

### TASL stability-based chemical screen identifies C5 as TLR7/8 inhibiting compound

We decided to start a small-molecule discovery campaign exploiting these two features: (i) a potentially druggable binding interface essential for complex formation and (ii) the instability of unbound TASL. We took advantage of the proteostatic relationship between SLC15A4 and TASL to devise an image-based assay to monitor the interaction. We engineered a reporter construct encoding C-terminally GFP-tagged TASL, followed by the P2A self-cleaving peptide and mCherry (TASL-EmGFP-P2A-mCherry, TGC reporter), which allowed constitutive expression of both ORFs from a common promoter (Fig. 1e). Thus, the comparison of GFP to mCherry fluorescence enables ratiometric monitoring of TASL-GFP proteostatic stability. After generating HEK293T cells stably expressing this TGC reporter, we observed that co-expression of SLC15A4 stabilized TASL-GFP protein, which is otherwise rapidly degraded (Fig. S1b). Confocal fluorescence microscopy analysis showed that TASL-GFP localized to lysosomes in presence of wildtype SLC15A4, while co-expression of a mutant lacking the N-terminal lysosomal targeting signal (ΔN-SLC15A4) redirected the TASL-GFP signal to the plasma membrane (Fig. 1e)[23]. This confirmed that in our TGC reporter system, TASL-GFP stabilization and localization was dependent on assembly with SLC15A4. For the chemical screening we selected a TGC reporter cell clone stably expressing SLC15A4 which showed a clear reduction of the TASL-GFP/mCherry ratio upon knockdown of SLC15A4 (Fig. 1f, g and Fig. S1c). To validate the phenotypic screening set-up and ascertain the feasibility of scoring complex assembly, we assessed whether SLC15A4-bound TASL-GFP could be displaced by defined N-terminal TASL portions. Transient overexpression of full-length C-terminally V5-tagged TASL (TASL-V5) in SLC15A4-TGC reporter cells resulted in reduced TASL-GFP levels, suggesting successful competition for SLC15A4 binding (Fig. S1d). Similar reduction was observed also in cells expressing high levels of N-terminal TASL fragments, with the construct encoding the first eight residues of TASL showing a significant effect. These experiments indicated that interfering with the SLC15A4-TASL interaction could reduce complex formation to a level detectable in our assay.

Based on these results, we used this cellular reporter system to screen a diversity library of 86.727 small molecules for their ability to specifically reduce TASL-GFP fluorescence without affecting mCherry control signal (Fig. 1h, i, Table S1). Reporter cells were incubated with compounds for 48 h before imaging as we aimed at monitoring degradation of the target protein, a process requiring time. mCherry signal was used as the basis for the detection of the cell area and images were evaluated for cell number and cell roundness to exclude highly toxic compounds and artifacts. Finally, the percentage of control (POC) normalized to cells treated with DMSO was evaluated with respect to GFP/mCherry ratio. Evaluation of the GFP signal was used as a filtering step to remove artifacts detected by the pipeline. 154 compounds that both decreased the GFP/mCherry ratio by >15% and reduced total GFP signal ( > 5% reduction) were identified as hits by the image analysis pipeline (Fig. 1j). After visual inspection we selected 12 compounds for further analysis. First, we monitored their effect on endogenous TASL protein in the TLR7/8-responsive human pDC cell line CAL-1 and observed that the different compounds reduced TASL levels to various degrees already after 24 h of pre-treatment (Fig. S2a). A salient feature of the SLC15A4-TASL axis is to specifically mediate IRF5 activation, while being dispensable for TLR7-9-induced NF-κB and MAPK activation[23]. We therefore decided to take advantage of this stringent specificity criterion to identify those compounds that would impair only IRF5. We developed a robust microscopy-based assay to monitor nuclear translocation of IRF5 and NF-κB member p65/RelA upon stimulation with TLR7/8 agonist R848 in CAL-1 cells (Fig. S2b, c). We first confirmed that SLC15A4 and TASL knockout selectively affected IRF5 but not p65 translocation (Fig. S2d) and tested the 12

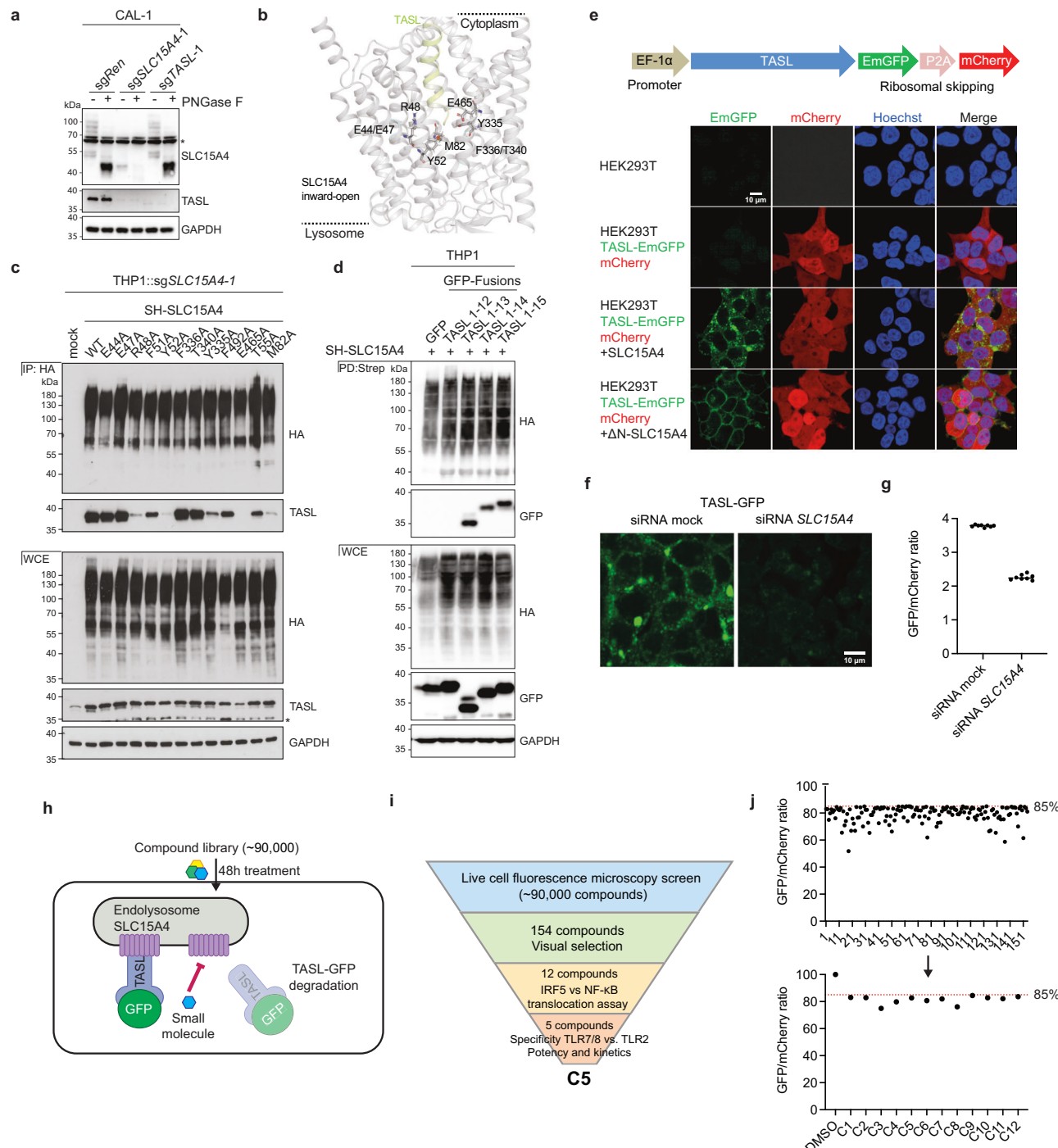

**Fig. 1 | SLC15A4-TASL binding interface and drug screening for TASL stability.**
**a** Immunoblot of whole cell extracts (WCE) from sg*SLC15A4*, sg*TASL* or control sg*Ren* CAL-1 cells, treated with PNGase F as indicated. Asterisk denotes a non-specific band. **b** AlphaFold model of SLC15A4 in complex with amino terminus of TASL (residues 1-20). **c** Immunoblots show WCE and HA immunoprecipitates (IP) from sg*SLC15A4*-expressing THP1 cells reconstituted with the indicated StepHA-tagged wildtype (WT) or mutant SLC15A4 variants. **d** Immunoblots show WCE and StrepTactin pull-downs (PD) from StrepHA-SLC15A4-expressing THP1 cells co-expressing the indicated GFP-fusion constructs. **e** Schematic of the TGC vector (top) and confocal live cell microscopy images of HEK293T stably

expressing the TGC vector, SLC15A4 and/or its plasma membrane localized variant ΔN-SLC15A4 as indicated (bottom). **f, g** Live cell confocal microscopy images (**f**) and GFP/mCherry fluorescence ratios (**g**) of the TGC reporter clone 48 h after transfection with the indicated siRNA. Bar shows mean, *n* = 8 biological replicates. **h** Schematic of screening principle to identify TASL-destabilizing compounds. **i** Summary of compound evaluation steps and validation assays leading to C5 selection. **j** POC GFP/mCherry ratios of 154 compounds identified by the image analysis pipeline (top) and of 12 compounds selected after visual inspection (bottom). **e, f** Scale bars, 10 μm. In **a, c**–**g** data are representative of at least two independent experiments.

candidates for their specificity. Five compounds (C2, C4, C5, C8 and C10) displayed specific inhibition of IRF5 but left the NF-κB pathway unscathed (Fig. 2a). Next, we used THP1 cells bearing an ISRE-driven reporter to investigate the kinetic profile as well as selectivity in

inhibiting responses induced by agonists of endolysosomal versus plasma membrane TLRs (Fig. S2e). Based on its potency and kinetics, on top of its specificity, we selected the compound C5 for further characterization (Fig. 2b). C5 is an elongated molecule comprised of an

aromatic 2-(4-ethoxyphenyl)-quinoline moiety followed by an amide bond and an aliphatic 2-methyl-1-propylpiperidine (Fig. 2c). To validate the scaffold and explore the chemical space at the aliphatic end of the compound, we obtained five analogs and a newly synthesized batch of C5 (Fig. S3a). All five analogs displayed comparable level of activity in the THP1 ISRE reporter assay, establishing that this activity is mediated by the common scaffold (Fig. S3b).

Finally, C5 inhibited TLR8-dependent responses to ssRNA40, confirming data obtained with R848, while it was not or less active on STING, NOD1/2 or RIG-like helicase pathways (Fig. S3c).

We therefore selected the C5 compound to assess effects on TLR7/8-induced signaling and responses.

## C5 inhibits IRF5-dependent responses

As the screening monitored interference with the TASL-SLC15A4 complex, SLC15A4 levels could also be affected. Thus, we assessed the endogenous level of SLC15A4 upon C5 treatment. In both CAL-1 and THP1 cells, C5 treatment resulted in dose-dependent reduction of TASL protein levels, while no significant effect was observed on endogenous SLC15A4 (nor on lysosomal protein LAMP1) (Fig. 2d, Fig. S4a). TASL loss was detectable as early as 2 h after C5 treatment and gradually increased (Fig. S4b). Upon stimulation with R848, C5 strongly impaired IRF5 activation in both cell types, without significantly affecting the NF-κB nor MAPK pathways (Fig. 2e, f, Fig. S4c, d). This further demonstrated that C5 does not influence TLR7/8 engagement by R848 or proximal signaling, and faithfully phenocopied SLC15A4/TASL deficiency. Key to any biological effect of C5 would be its ability to modulate proinflammatory cytokines and type I interferon (IFN) production. C5 strongly suppressed R848-induced responses in CAL-1 pDCs, and a similar response was observed in THP1 cells (Fig. 2g, h, Fig. S4e, f). In line with the specificity profile determined in reporter cells, C5 strongly inhibited the production of TNF and CCL5 by THP1 cells upon R848 treatment, while it only had modest (Pam3CSK4/TLR2) or no inhibitory effects (LPS/TLR4, cGAMP/STING) on the other tested pattern recognition pathways (Fig. 2i, Fig. S4g). Altogether these data demonstrate that C5 specifically impairs TLR7/8-induced IRF5 activation as well as downstream cytokines/chemokines and type I IFN production in human monocytic and pDC cell lines.

To assess activity in human primary cells specifically relevant for SLE pathophysiology, we first monitored the effect of C5 in peripheral blood mononuclear cells (PBMCs) from healthy donors. In line with suppressed IRF5-dependent signaling, C5 potently impaired TNF and to a lesser extent IL-6 production upon R848 treatment of bulk PBMCs (Fig. 3a). We had previously shown that the SLC15A4-TASL module is required for TNF production in primary human monocytes[23]. We purified CD14[+] cells and were able to show that C5 caused a significant reduction in this proinflammatory cytokine (Fig. 3b). Among endolysosomal TLR responsive cells, B cells play a central role in SLE pathogenesis and are the targets of current therapeutics[9,10,29]. We therefore assessed the impact of C5 on TLR7/8 responses by directly monitoring IRF5 in B cells by confocal microscopy. Remarkably, C5 significantly reduced IRF5 nuclear translocation, without affecting the activation of NF-κB p65 (Fig. 3c, d, Figs. S5 and S6). These data demonstrate that C5 activity extends to disease-relevant human primary immune cells and highlight the exquisite selectivity for targeting the IRF5 signaling branch also in primary human B cells. Next, we obtained PBMCs from SLE patients. While unstimulated cells did not show detectable levels of TNF, they responded strongly to stimulation by R848 (Fig. 3e). Treatment with C5 completely abolished stimulus-dependent TNF production. Again, we also monitored selective IRF5 blockade by C5 in the nuclear translocation assay and, as observed in healthy donors, the compound inhibited IRF5 but not NF-κB p65 (Fig. 3f). Together, these experiments positively assessed the activity of the compound in disease-relevant settings.

## The mechanism of action of C5 requires SLC15A4

Encouraged by the fact that the chemical entity was showing the desired behavior and specificity profile, it became important to better clarify the molecular basis of this uncommon mechanism of action, i.e. interference with a highly specific protein interaction leading to degradation of a single complex component. We first monitored the effect of C5 on complex assembly. As expected, compound treatment reduced TASL-GFP levels in HEK293T cells co-expressing SLC15A4 (Fig. 3g, Fig. S7a). In contrast, C5 had only a minor effect on total levels of GFP-SLC15A4 protein when expressed alone. When monitoring this effect by microscopy, we indeed observed a reduction of TASL-GFP levels accompanied by relocalization (Fig. 3h, Fig. S7b). While total levels of GFP-SLC15A4 were unaltered, localization appears to reflect some mild changes in lysosomal morphology. To untangle the TASL degrading activity from any possible lysosomal effects, we took advantage of our ability to artificially localize the complex to the plasma membrane using ΔN-SLC15A4. C5 did not affect levels or localization of GFP-ΔN-SLC15A4 (Fig. 3g, h, Fig. S7a, b). Importantly, in cells co-expressing ΔN-SLC15A4 and TASL-GFP, C5 treatment severely reduced plasma membrane-associated TASL levels (Fig. 3g, h, Fig. S7a, b). This strongly suggests that C5 is acting on the TASL-SLC15A4 complex directly, as it affects TASL levels even when the complex is experimentally localized at the plasma membrane, excluding therefore a specific requirement for the lysosomal environment. Next, we mapped the requirement for C5 activity on the complex and monitored its effect on the association between GFP-SLC15A4 and transiently transfected V5-tagged TASL constructs encoding the N-terminal 94 or 193 amino acids. Immunofluorescence analysis revealed that C5 reduced total and SLC15A4-colocalized V5 signal for both TASL fragments (Fig. S7c). To further address the possibility that the compound would act downstream of SLC15A4 and possibly directly on TASL, we relied on another instance of engineered protein localization. We recently found that fusion of TASL to the lysosomal targeting sequence of LAMTOR1 (encoded by amino acids 1-81) bypasses the requirement of SLC15A4 for TLR7/8-induced IRF5 activation[30]. Indeed, expression of the LAMTOR1(1:81)-TASL fusion protein in SLC15A4-deficient CAL-1 cells efficiently rescued IRF5 activation (Fig. 3i). C5 treatment failed to impair IRF5 activation mediated by this lysosomal localized TASL, while showing the expected inhibition in control (sgRen) cells which relies on the endogenous SLC15A4-TASL complex (Fig. 3i). In line with this, C5 impaired SLC15A4-dependent IL-6 production in control cells, but had no effect on cytokine production in SLC15A4 knockout cells expressing LAMTOR1(1:81)-TASL (Fig. 3j). Failure of C5 to impair IRF5 activation and downstream responses of a pathway engineered to bypass SLC15A4 provides experimental evidence for SLC15A4 representing the primary target of C5.

## C5 locks SLC15A4 in a TASL binding-incompetent lysosomal outward-open conformation

Thermal shift experiments using cell lysates as well as competition experiments with TASL-SLC15A4 co-precipitates failed to provide evidence for SLC15A4 binding by C5, challenging the notion that C5 would compete directly for TASL engagement. As this contrasted with all the data obtained with intact cells, it suggested a mechanism of action resulting in TASL disengagement that was not driven by simple competition for binding. With the availability of the structure of human SLC15A4, both in its lysosomal outward-open and the TASL-bound inward-open conformations, came the possibility to directly assess the C5 binding mode (see ref. 31).

We used cryo-electron microscopy (cryo-EM) to determine the structure of C5 in complex with the SLC15A4/ALFA_Nanobody (Fig. S8). After purification of the SLC15A4/C5 complex and cryo-EM analysis, the 2D averages of the SLC15A4/C5 complex displayed both the dimeric and monomeric conformations, similar with the apo

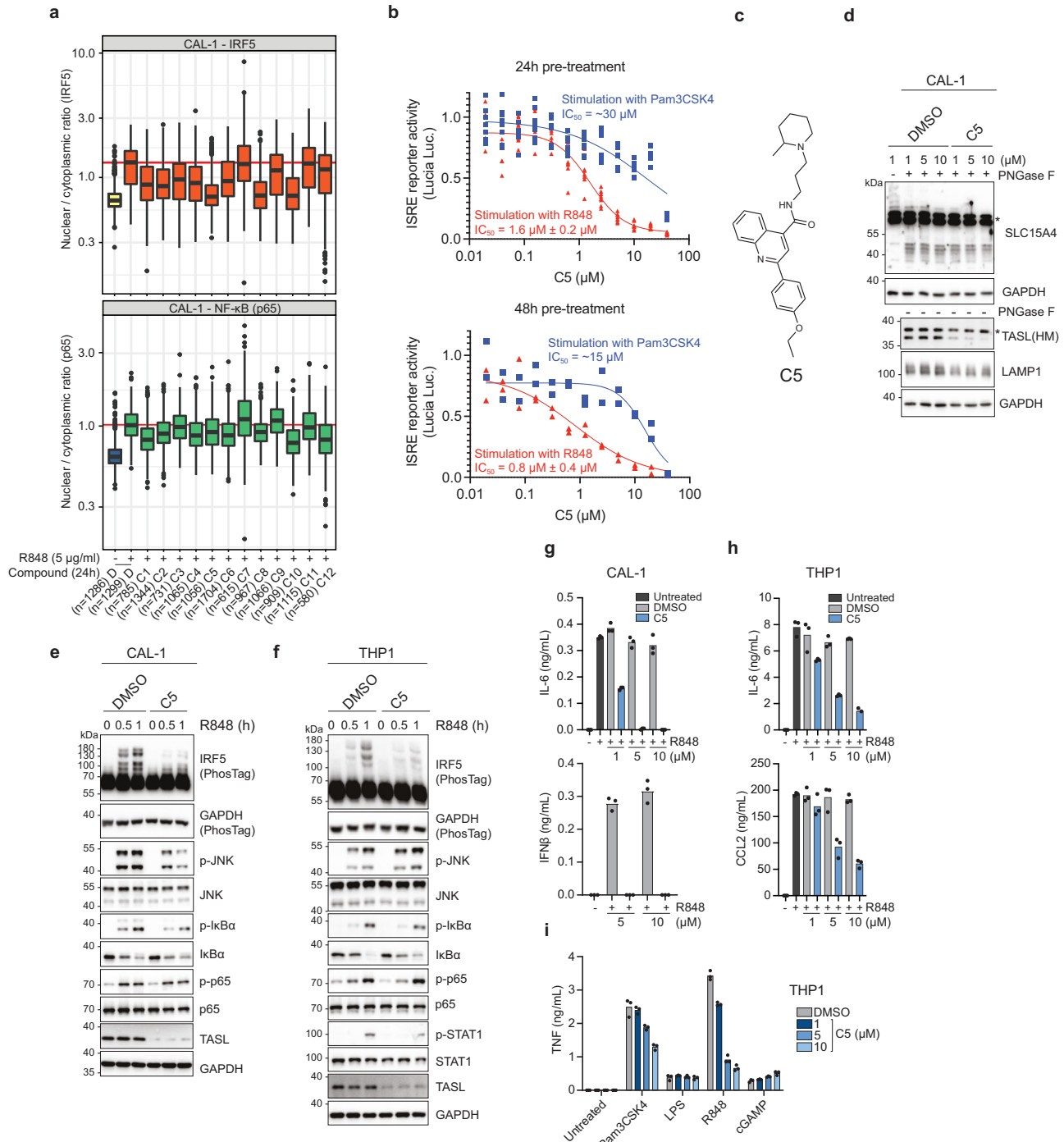

**Fig. 2 | Identification of C5 as an IRF5 pathway-specific inhibitor. a** Image-based evaluation of IRF5 (top) and NF-κB (bottom) nuclear translocation in CAL-1 cells. Cells were pre-treated for 24 h with the indicated compounds (10 μM) or DMSO, stimulated with R848 (5 μg/ml) for 3 h as indicated and analyzed by confocal microscopy. Bars indicate the median, boxes indicate the first to third quartiles. The top whisker extends from hinge to largest value no further than 1.5 × interquartile range (IQR) from the hinge, and the bottom whisker extends from the hinge to smallest value at most 1.5 × IQR of the hinge. Dots indicate outlying points. *n* (cells) as indicated. D, DMSO. **b** Dose response of C5 in THP1 DUAL reporter cells. Cells were pre-treated for 24 h (top) and 48 h (bottom) before stimulation with R848 (5 μg/ml, red triangles) or Pam3CSK4 (0.1 μg/ml, blue squares) for 20 h. Data show (top panel, 24 h) *n* = 6 biological replicates and (bottom panel, 48 h) *n* = 2 biological

replicates. Representative of three independent experiments. **c** Chemical structure of C5. **d** Immunoblots of CAL-1 upon 24 h treatment with C5 or vehicle DMSO. Lysates were treated with PNGase F as indicated. Asterisk denotes non-specific bands. (**e, f**) Immunoblots of CAL-1 (**e**) or THP1 (**f**) cells pre-treated with C5 (10 μM) for 24 h before R848 (5 μg/ml) treatment for the indicated time. **g**–**i** Supernatants from CAL-1 (**g**) or THP1 (**h, i**) cells pre-treated with C5 for 24 h before stimulation (5 μg/ml R848, 0.1 μg/ml Pam3CSK4, 0.1 μg/ml LPS, 3 μg/ml cGAMP for 16-24 h) were analyzed by ELISA for the indicated cytokines. Data show mean of three stimulation replicates from one experiment representative of two independent experiments. In (**a, b, d**–**f**) data are representative of at least two independent experiments.

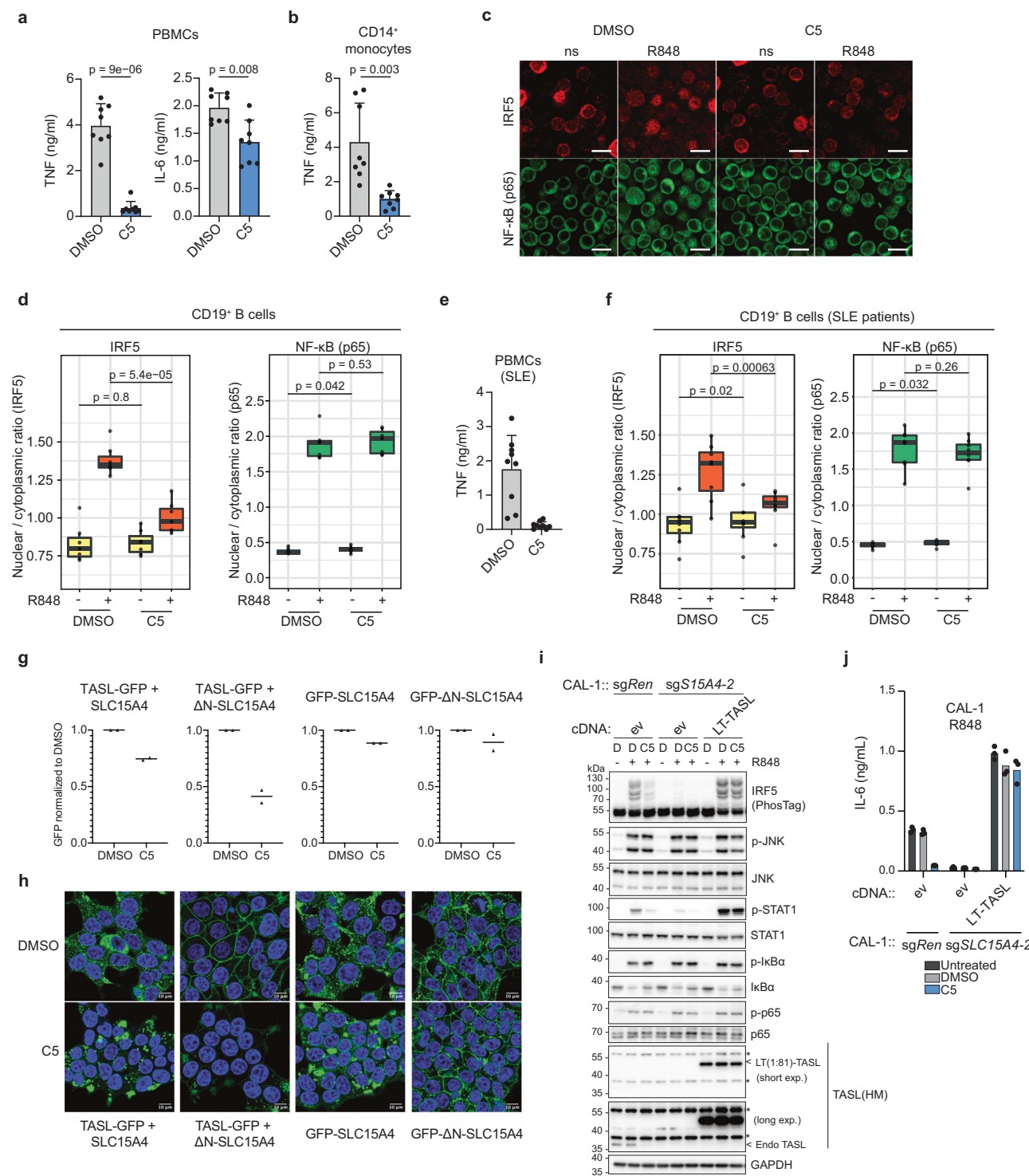

SLC15A4 (see accompanying paper by Chen, Xie, Zhang et al.). Finally, we resolved the structure of dimeric SLC15A4/C5 complex at an overall resolution of 2.50 Å (Fig. 4a, b). The cryo-EM map showed well-resolved densities and was sufficient to successfully build most of the protein region, the C5 molecule and two cholesterol molecules per protomer at the dimer interface (Fig. S9). The dimeric SLC15A4/C5 complex adopts an outward-open conformation and surface electrostatics analysis revealed that the outward-open cavity is negatively charged (Fig. 4c, d). We found an extra density at the bottom of this cavity which was identified as a C5 molecule in each protomer of the SLC15A4/C5 complex (Fig. S10a–c). C5 binding occurs at the canonical substrate binding site within the large family of proton-coupled

oligopeptide (POT) transporters, where dipeptides are seen interacting in the human SLC15A1 and SLC15A2 cryo-EM structures (Fig. S10d)[28,32]. Assisted by the high-resolution map, we finally determined that the bound C5 molecule within the cavity adopts an L-shaped binding pose, which was coordinated by several interactions at the interface of the N- and C-bundles, including electrostatic and hydrophobic interactions (Fig. 4e, f and Fig. S10e). Residues involved in the interaction between SLC15A4 and C5 are highly conserved within vertebrates (Fig. S11). Structural comparison between the SLC15A4/C5 complex and the apo SLC15A4 showed no obvious movement of the TMs, with a root mean square deviation (RMSD) of 0.394 over 878 Cα atoms (Fig. 4g). SLC15A4 did not undergo a

**Fig. 3 | C5 acts through SLC15A4 to selectively inhibit TLR7/8-induced IRF5 responses in cells from healthy donors and SLE patients. a** PBMCs or **b** isolated CD14[+] monocytes from healthy donors were pre-treated for 24 h with C5 (10 μM) and stimulated for 16 h with R848 (5 μg/ml). Supernatants were analyzed by ELISA as indicated. Data show mean ± SD, $n$ = 8 donors. **c** Representative confocal microscopy images of PBMCs pre-treated for 24 h with C5 (10 μM) and stimulated for 3 h with R848 (5 μg/ml) stained for IRF5 and NF-κB p65. **d** PBMCs from healthy donors pre-treated for 24 h with DMSO or C5 (10 μM) and stimulated with R848 (5 μg/ml) for 3 h as indicated were analyzed for CD19[+] B cell-specific IRF5 and NF-κB p65 nuclear translocation. $n$ = 7 donors. **e** PBMCs from SLE patients were pre-treated for 24 h with C5 (10 μM) and stimulated for 16 h with R848 (5 μg/ml). Supernatants were analyzed by ELISA as indicated. Data show mean ± SD, $n$ = 9 donors. **f** PBMCs of SLE patients pre-treated for 24 h with DMSO or C5 (10 μM) and stimulated with R848 (5 μg/ml) for 3 h as indicated were analyzed for CD19[+] B cell-specific IRF5 and NF-κB p65 nuclear translocation. $n$ = 8 donors. **g, h** Flow cyto-metry analysis (**g**) or live cell fluorescence microscopy (**h**) of HEK293T cells stably expressing the indicated constructs after 24 h treatment with DMSO or 10 μM C5. **g** Mean of integrated GFP signals from two independent experiments shown. See Supplementary Note 2 for gating strategy. **i** Immunoblot of control sg*Ren* or sg*SLC15A4* CAL-1 cells stably reconstituted with indicated constructs. Cells were pre-treated with C5 (10 μM) for 24 h before R848 (5 μg/ml) treatment for the indicated time. **j** Supernatants from indicated CAL-1 cell lines pre-treated with C5 for 24 h before R848 (5 μg/ml, 24 h) stimulation were analyzed for IL-6 production by ELISA. Data show mean of three stimulation replicates from one experiment representative of two independent experiments. (**a**, **b**, **d**, **f**) Significance was assessed by paired two-sided *t-test*. (**c**, **h**) Scale bars, 10 μm. (**d**, **f**) Box plots defined as in Fig. 2a. In (**h**–**j**) data representative of at least two independent experiments.

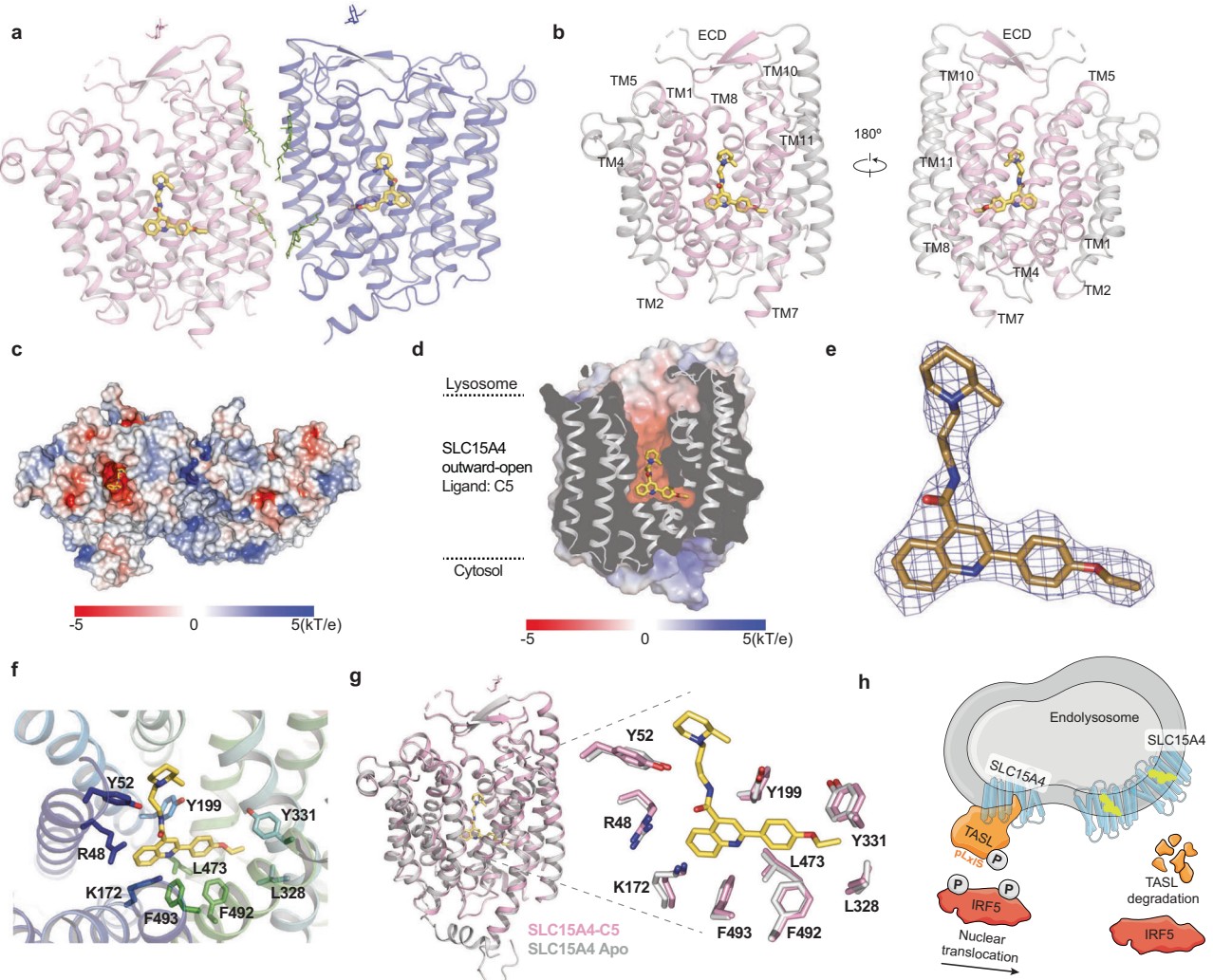

**Fig. 4 | C5 locks SLC15A4 in an outward-open, TASL binding-incompetent conformation. a, b** Ribbon model of human SLC15A4/C5 complex. In **a** the two protomers of SLC15A4 are colored pink and blue, respectively. The bound C5 molecules are shown as sticks and colored dark yellow. In **b** the TMs surrounding C5 are highlighted. **c, d** Heatmap of the surface electrostatics of human SLC15A4 in the outward-open C5-bound state showing the C5 binding pocket. The C5 molecule is shown as dark yellow sticks. **e** Cryo-EM density showing the bound C5 in SLC15A4/C5 complex (blue mesh, 6σ). The C5 molecule is shown as sticks. **f** Close-up view of the C5 binding site in SLC15A4/C5 complex. **g** Structural comparison between the SLC15A4/C5 complex (pink) and the SLC15A4 in the apo state (gray), only one protomer is shown. **h** Schematic model summarizing the mode-of-action of feeblin/C5. Feeblin/C5 is indicated in yellow.

significant conformational change after binding with C5 and only subtle movement of the sidechains of the C5 binding pocket residues was observed (Fig. 4g). Collectively, the structural data reveal that C5 stabilizes SLC15A4 in its lysosomal outward-open conformation. The cryo-EM structure of the TASL N-terminal portion bound to SLC15A4 (described in Chen, Xie, Zhang et al.) shows an inward-open conformation, that starkly contrasts the C5-bound form. This is enlightening regarding the regulatory mechanisms behind the opposite effects of C5 and TASL engagement, which stabilize the outward or inward-open state respectively.

## Discussion

This study presents the discovery of a compound with an unusual mechanism of action: stabilization of a conformational switch within an MFS fold protein (SLC15A4) that is incompatible with interaction with an effector protein (TASL), which is rapidly degraded when unbound (Fig. 4h). This results in the physical interruption of a disease-associated proinflammatory signaling pathway evolved to convey detection of pathogen-associated molecular patterns by TLR7/8 in the lysosome to the IRF5 transcription factor and its target genes. Interruption of this pathway is highly effective, as TASL represents the physical bridge between SLC15A4 and IRF5. The pathway-specificity and conformation-selectivity of the effect stems from the exploitation of this natural, proteostatic relationship able to turn a reversible event, such as drug engagement, to an irreversible, such as enzymatic degradation of the target effector protein. As such, while the mechanism of action resembles inhibition of a transporter protein, it functions as an allosteric inducer of targeted protein degradation. It is worth observing that a compound with such a sophisticated proteostatic mechanism is the result of a phenotypic screen on a focused read-out, and not of design. C5 was obtained directly out of a primary screening campaign and, without any optimization, displayed activity at low μM concentrations on both TASL stability and TLR7/8-induced proinflammatory responses. At these concentrations, C5 manifests specificity for endolysosomal TLRs, without major effects on other innate immune pathways, including TLR2, TLR4, STING, and NOD1/2 signaling. Nevertheless, enabled by the SLC15A4/C5 cryo-EM structure presented here, future structure-activity relationship optimization efforts will aim at increasing potency and possibly selectivity as at concentrations above 10 μM, C5 displayed a partial inhibitory effect on Pam3CSK4-induced responses as well as a mild synergy with STING agonists, possibly reflecting the observed remodeling effect on lysosomes. Interestingly, C5 seems to exhibit stronger activity on the SLC15A4-TASL complex when localized to the plasma membrane as compared to the lysosome. This could reflect increased accessibility of the SLC15A4 binding pocket and/or higher availability of the compound in the extracellular milieu than in the lysosomal lumen, which could possibly be influenced by lysosomal uptake or efflux mechanisms, or compound metabolization.

SLC15A4 displays significant homology to SLC15A3, the other SLC15A family member with lysosomal localization. While current data cannot exclude possible effects of C5 on SLC15A3, multiple lines of evidence strongly support that the observed inhibitory effect on TLR7-9 responses is mediated by targeting SLC15A4 and not SLC15A3. Indeed, we have previously shown that SLC15A3 knockout does not affect TLR7-9 responses in THP1 cells, expression of SLC15A3 fails to rescue responses in SLC15A4-deficient cells and SLC15A3 does not bind TASL[23]. Moreover, in contrast to SLC15A4, SLC15A3 is not required for endosomal TLR9 responses in primary murine pDCs[33]. Lastly, the observation that C5 did not inhibit TLR7/8 responses in SLC15A4-deficient cells in which TASL was constitutively tethered to lysosomes (Fig. 3i, j) demonstrates that its activity is strictly mediated through SLC15A4 and not by any other possible off-targets.

Considering the exquisite nature of the allosteric regulation exploited by C5, it is tempting to speculate that this may reflect a physiological mechanism involving an endogenous lysosomal metabolite, which could modulate and time the pathway output. Our recent finding that lysosomal tethering of TASL is sufficient to rescue TLR7-9 responses in SLC15A4-deficient cells demonstrates that SLC15A4 transport activity is not in itself required for activation of this pathway[30]. However, we do not exclude that binding and/or transport of substrates (such as the previously proposed oligopeptides or histidine, or others, which might still be uncharacterized) could regulate the SLC15A4-TASL complex assembly. Indeed, the SLC15A4-TASL binding mode described by Chen, Xie, Zhang et al. in the accompanying paper strongly suggests that any molecule stabilizing SLC15A4 in the outward or inward conformation is expected to inhibit or promote TASL recruitment, respectively. Future investigations will aim at the identification of possible natural ligands, possibly making use of C5 in competition experiments and as structural guide.

As C5 targets SLC15A4, and effectively mimics its loss of function, we propose to name the compound feeblin after *feeble*, the first described loss of function mutant in mice[14]. A high number of SLE patients insufficiently respond or are refractory to currently available treatments, highlighting the need for development of more precise medicines exploiting disease-specific processes[29,34]. With the evident genetic association of the SLC15A4/TASL/IRF5 module with lupus, feeblin represents an encouraging strategy for targeting an etiological mechanism. Whether feeblin itself will represent the starting point of a future therapeutic or not, it provides a mechanistic proof-of-principle for the development of a potential anti-inflammatory drug for SLE and related diseases.

## Methods

### Ethical statement

Human primary PBMCs were isolated from the blood of anonymous healthy donors and SLE patients included in our biobank approved by the Ethics Committee of the Medical University of Vienna (EK2071/2020 and EK1075/2021). All donors provided written informed consent and study procedures follow good clinical practice guidelines and the Declaration of Helsinki.

### Antibodies and reagents

Antibodies used in western blot: Rabbit anti-TASL (Sigma, Cat.#: HPA001185, Lot: 000030856, Dilution 1:1000), rabbit anti-IRF5 (Abcam, Cat.#: ab181553, Clone: EPR17067, Lot: GR3248905-4, Dilution 1:1000), mouse anti-LAMP1 (Santa Cruz, Cat.#: sc-20011, Clone: H4A3, Lot: H0321, Dilution 1:1000), mouse anti-GAPDH (Santa Cruz, Cat.#: sc-365062, Clone: G-9, Lot: G0121 and I2321, Dilution 1:1000), mouse anti-IκBα (Cell Signaling, Cat.#: 4814, Clone: L35A5, Lot: 17, Dilution 1:1000), rabbit anti-phospho-IκBα Ser32 (Cell Signaling, Cat.#: 2859, Clone: 14D4, Lot: 18, Dilution 1:1000), rabbit anti-SAPK/JNK (Cell Signaling, Cat.#: 9252, Lot: 17, Dilution 1:1000), rabbit anti-phospho-SAPK/JNK Thr183/Tyr185 (Cell Signaling, Cat.#: 4668, Lot: 15, Clone: 81E11, Dilution 1:1000), rabbit anti-STAT1 (Cell Signaling, Cat.#: 14994, Clone: D1K9Y, Lot: 5, Dilution 1:1000), rabbit anti-phospho-STAT1 Tyr701 (Cell Signaling, Cat.#: 7649, Clone: D4A7, Lot: 5, Dilution 1:1000), rabbit anti-NF-κB p65 (Cell Signaling, Cat.#: 8242, Clone: D14E12, Lot: 16, Dilution 1:1000), rabbit anti-phospho-NF-κB p65 Ser536 (Cell Signaling, Cat.#: 3033, Clone: 93H1, Lot: 17, Dilution 1:1000), rabbit anti-HA (Cell Signaling, Cat.#: 3724, Clone: C29F4, Lot: 10, Dilution 1:1000), rabbit anti-GFP (Cell Signaling, Cat.#: 2956, Clone: D5.1, Lot: 6, Dilution 1:1000). Custom rabbit anti-SLC15A4 antibodies were generated by Genscript (raised against the N-terminus of SLC15A4, Dilution 1:1000)[23]. Custom rabbit anti-TASL antibodies (TASL HM) were produced by Eurogentec (recognizing the C-terminus of TASL, Dilution 1:1000)[30]. Antibodies used in immunofluorescence: Rabbit anti-IRF5-Alexa647 (Cell Signaling, Cat.#: 74818, Clone: E7F9W, Lot: 1, Dilution 1:200), rabbit anti-NF-κB p65-Alexa488 (Cell Signaling, Cat.#: 49445, Clone: D14E12, Lot: 3, Dilution 1:200), mouse anti-CD19-PE (BioLegend, Cat.#: 302254, Clone: HIB19, Lot: B325450, Dilution 1:200), mouse anti-V5 (Invitrogen, Cat.#: R960-25, Clone: SV5-Pk1, Lot: 2378586, Dilution 1:1000), goat anti-mouse Alexa-568 (Invitrogen, Cat.#: A11004, Lot: 2090670, Dilution 1:500). Antibodies used in flow cytometry: Rabbit anti-V5 (Cell Signaling, Cat.#: 13202, Clone: D3H8Q, Lot: 6, Dilution 1:1000), goat anti-rabbit IgG (H + L), F(ab') 2 fragment-Alexa647 (Cell Signaling, Cat.#: 4414, Dilution 1:1000). Specificity of custom rabbit anti-SLC15A4 (Genscript) and anti-TASL (TASL HM, Eurogentec) have been validated previously[23,30]. All other antibodies were bought from commercial vendors and validation for indicated species and applications can be found on the manufacturers website or the provided scientific citations on the same website. R848 (tlrl-r848), Pam3CKS4 (tlrl-pms), LPS (tlrl-3pelps), cGAMP

(tlrl-nacga23), ssRNA40/LyoVec (tlrl-lrna40), Poly(I:C) (tlrl-pic), C12-iE-DAP (tlrl-c12dap) and L18-MDP (tlrl-lmdp) were from Invivogen. C5 was purchased from ChemDiv Inc and Enamine and C5 analogs were purchased from ChemDiv Inc. Other early lead molecules were acquired from Vitas M Chemical Limited (C1), ChemBridge (C8), and Chemspace LLC (C10 and C4). Purity and identity of C5 was confirmed by NMR and mass spectrometry by Enamine (Supplementary Note 1).

## Cell culture and human primary cells

HEK293T cells (Cat. #: CRL-3216) and THP1 cells (Cat. #: TIB-202) were purchased from ATCC and authenticated by short tandem repeat profiling. THP1 DUAL reporter cell lines (Cat. #: dhpd-nfis) were obtained from Invivogen. CAL-1 cells were provided by T. Maeda (Nagasaki University). Cell lines were regularly tested for mycoplasma contamination. Human primary PBMCs were isolated from the blood of anonymous healthy donors and SLE patients. CD14$^+$ monocytes were purified by magnetic-activated cell sorting (Miltenyi Biotec, #130-050-201) according to the manufacturers protocol. Purity of CD14$^+$ cells ranged between 80.8% and 94.2% as determined by flow cytometry analysis. HEK293T cells were cultured in DMEM, THP1, CAL-1, primary human PBMCs and CD14$^+$ monocytes in RPMI, supplemented with 10% (v/v) FBS and antibiotics (100 U/ml penicillin, 100 μg/ml streptomycin), all from Gibco. Cells were incubated at 37 °C in 5% CO2.

## Plasmids and siRNAs

Codon-optimized cDNAs for human wildtype and mutant SLC15A4 and TASL were obtained from Genscript and subcloned into pDONR201 (Invitrogen) plasmids as reported previously[23]. The vector required for creation of the TGC cell line was assembled from a gateway (gw) compatible pRRL-EF1-gw-EmGFP-IRES-HygroR vector and a vector containing the P2A-mCherry sequence. The human codon-optimized TASL sequence was then subcloned into the resultant gateway compatible vector. Human codon-optimized sequences of SLC15A4 and ΔN-SLC15A4 (29-end) were cloned into a pRRL-EF1a-EmGFP-gw-IRES-HygroR backbone with the GFP-fusion protein at the amino terminus.

For lentiviral transduction, cDNAs were shuttled to pRRL-based lentiviral expression plasmids and a previously described selectable resistance cassette[35]. Lentiviral packaging plasmids psPAX2 and pMD2.G were obtained from Addgene (Plasmid # 12260 and 12259, kind gift from Didier Trono). Full-length TASL and shorter constructs (residues 1-8, 1-15, 1-94 and 1-193) were cloned into a pDONR221 vector without a stop codon and further subcloned into a pLIX403 plasmid with a C-terminal V5-tag (Addgene Plasmid # 41395, kind gift from David Root). Fusions of TASL N-terminal sequences (aa 1-12, 1-13, 1-14 and 1-15) with GFP were generated by introducing corresponding annealed oligos into the BsmBI sites of the Artichoke reporter vector (Addgene Plasmid # 73320, kind gift from Benjamin Ebert) via Golden Gate assembly. The used oligo sequences were as follows: TASL 1-12 F: 5′- ccatGCTGAGCGAGGGCTATCTGAGCGGACTGGAGTAT-3′, TASL 1-12 R: 5′- cagcATACTCCAGTCCGCTCAGATAGCCCTCGCTCAGC-3′; TASL 1-13 F: 5′- ccatGCTGAGCGAGGGCTATCTGAGCGGACTGGAGTATTGG-3′, TASL 1-13 R: 5′- cagcCCAATACTCCAGTCCGCTCAGATAGCCCTCGCTCAGC-3′; TASL 1-14 F: 5′- ccatGCTGAGCGAGGGCTATCTGAGCGGACTGGAGTATTGGAAC-3′, TASL 1-14 R: 5′- cagcGTTCCAATACTCCAGTCCGCTCAGATAGCCCTCGCTCAGC-3′; TASL 1-15 F: 5′-ccatGCTGAGCGAGGGCTATCTGAGCGGACTGGAGTATTGGAACGAC-3′, TASL 1-15 R: 5′-cagcGTCGTTCCAATACTCCAGTCCGCTCAGATAGCCCTCGCTCAGC-3′. LAMTOR1(1:81)-TASL fusion was cloned by Gibson assembly into a pRRL-based lentiviral expression plasmid[30]. siRNAs targeting codon-optimized human SLC15A4 and a non-targeting Control Pool (cat.: D-001810-10-05) were obtained from Dharmacon. The sequences for the siRNAs were as follows (5′ to 3′ orientation):

- SLC14A siRNA no. 1: GGGCAGCCUUCUACGGAAUUU
- SLC15A4 siRNA no. 2: CAACCAGACCAUCGGCAAUUU

siRNA transfection was performed with a reverse transfection protocol using RNAiMX Lipofectamine (Thermo Fisher). sgRNA used have been previously described[23]: cloned oligonucleotides were as follows (5′ to 3′ orientation):

- SLC15A4 sgRNA no. 1, F: CACCGGGAGCGATCCTGTCGTTAGG, R: AAACCCTAACGACAGGATCGCTCCC
- SLC15A4 sgRNA no. 2, F: CACCGTATTACAACCACTCCTCACA, R: AAACTGTGAGGAGTGGTTGTAATAC
- TASL sgRNA no. 1, F: CACCGGTAGAAATGGAATCCTCCAT, R: AAACATGGAGGATTCCATTTCTACC
- TASL sgRNA no. 2, F: CACCGCTGAATTAATGGCCATCACC, R: AAACGGTGATGGCCATTAATTCAGC

## Lentiviral transduction

For lentiviral gene transduction, HEK293T cells were transfected with the respective lentiviral vectors and packaging plasmids psPAX2 and pMD2.G using PEI (Sigma). Twenty-four hours later, medium was exchanged to DMEM or RPMI, supplemented with 10% (v/v) FBS and antibiotics (100 U/ml penicillin, 100 μg/ml streptomycin). Seventy-two hours after transfection, cell supernatants were collected, filtered through 0.45 μm polyethersulfone filters (GE Healthcare) and supplemented with 8 μg/ml protamine sulfate (Sigma). Cells were infected with virus containing protamine sulfate supplemented supernatants. Twenty-four hours after infection, medium was changed; forty-eight hours after infection, cells were selected with the respective antibiotics.

## Compound treatment and R848 stimulation

CAL-1, THP1 or human primary cells were incubated with C5 and other identified compounds (or the corresponding volume of DMSO vehicle) for 24 h or 48 h at the indicated concentration and subsequently stimulated with R848 (5 μg/ml) for the indicated time. Supernatants or cells were recovered for ELISA, SDS–PAGE or microscopy-based analysis.

## Cell lysis and western blotting

Cells were lysed in RIPA lysis buffer (25 mM Tris, 150 mM NaCl, 0.5% NP-40, 0.5% deoxycholate (w/v) and 0.1% SDS (w/v), pH 7.4) supplemented with Roche EDTA-free protease inhibitor cocktail (one tablet for 50 mL) and halt phosphatase inhibitor cocktail (Thermo Fisher Scientific), 10 min on ice. After lysate clearing by centrifugation (16,000 g, 10 min, 4 °C), proteins were quantified with BCA (Thermo Fisher Scientific) using BSA as standard. Cell lysates were resolved by regular or Phos-Tag-containing (20 or 50 μM, WAKO Chemicals) SDS–PAGE. After electrophoresis, Phos-Tag-containing SDS–PAGE were soaked in transfer buffer with 10 mM EDTA for 3 × 10 min, rinsed 10 min in transfer buffer without EDTA and blotted to nitrocellulose membranes (Amersham, Glattbrugg, Switzerland). Membranes were blocked with 5% non-fat dry milk in TBST and probed with the indicated antibodies. Binding was detected with anti-mouse-HRP secondary antibodies (no.115-035-003) or anti-rabbit-HRP secondary antibodies (no.111-035-003), from Jackson Laboratories, using the ECL western blotting system (Advansta). When multiple antibodies were used, equal amounts of samples were loaded on multiple SDS–PAGE gels and western blots were sequentially probed with a maximum of two antibodies. Uncropped immunoblot figures are provided as source data file.

## Co-immunoprecipitation and StrepTactin pull-down

For co-immunoprecipitation and StrepTactin pull-downs, $1 \times 10^7$ THP1 cells were lysed in E1A (50 mM HEPES, 250 mM NaCl, 5 mM EDTA, 1% NP-40, pH 7.4) lysis buffer supplemented with Roche EDTA-free protease inhibitor cocktail (1 tablet per 50 ml) for 10 min on ice. Lysates were cleared by centrifugation for 10 min at 16.000 g, 4 °C and normalized using Bradford assay (Bio-Rad). After removal of whole cell

lysate as input, the remaining material was subjected to immunoprecipitation with anti-HA agarose (Sigma) or pull-down with StrepTactin beads (IBA Lifesciences) overnight at 4 °C. After washing of beads three times with E1A buffer, proteins were eluted with SDS sample buffer and analyzed by western blot.

### PNGase treatment

Cells were lysed in RIPA buffer. Per sample, 35 µl of cleared lysate was either incubated with or without 0.5–1 µl (500–1000 U) PNGase F (NEB) for 30 min at 37 °C. Samples were analyzed by western blotting.

### Enzyme-linked immunosorbent assay

All enzyme-linked immunosorbent assay (ELISA) experiments were performed using diluted cell supernatant according to manufacturer's instructions. ELISA kits for human IL-6 (no. 88-7066-88), human CCL2 (no. 88-7399-88) and human TNF (no. 88-7346-77) were from Invitrogen. ELISA kit for human interferon beta (no. 41410-1) was from PBL Assay Science, for human CCL5 (no. DY278) from R&D Systems.

### THP1 DUAL cell reporter assay

THP1 DUAL cells ($1 \times 10^5$ cells per 96 well) were incubated with compounds for 24 h or 48 h, and subsequently stimulated for 20–24 h with different ligands as indicated. Poly(I:C) was complexed with lipofectamine (1:1 ratio) (Invitrogen, 11668019) in Opti-MEM medium (Gibco) for 20 min before stimulation of cells. Cell culture supernatant were collected, cleared of residual cells by centrifugation and analyzed for NF-κB and ISRE reporter activity according to the manufacturer's instructions. For experiments in 384-well format, compounds were pre-plated at defined concentrations and THP1 DUAL cells were added at 32k/well concentration in 40 µL. After incubation for 24 h or 48 h cells were stimulated with 10 µL of a 25 µg/ml stock of R848 or 0.5 µg/xml Pam3CKS4 5x stocks as indicated for 20 h and detected by directly adding Quanti-Luc Plus (Invivogen) solution to the plates, according to the manufacturer's instructions and without adding stabilizer solution. Quanti-Blue detection was carried out according to the manufacturer's instructions, with QB reagent and buffer (Invivogen) dissolved in 30 ml water, instead of the 100 ml used for standard detection. Both assays were detected with an Envision 2104 Multilabel reader and results were analyzed in Prism version 9.

### Flow cytometry

HEK293T cells transfected with TASL-V5 plasmid were fixed and permeabilized with IC fixation and permeabilization buffer (Invitrogen) before staining for V5-tag and detected with deep red Alexa Fluor 647 secondary antibody to allow for selection in the presence of an mCherry reporter. Cells were measured either immediately after staining or in the case of live cells immediately after harvesting and washing with PBS. Data were acquired on a BD FACSCalibur (BD Biosciences) and analyzed with FlowJo software (version 10). Gating strategies are provided in Supplementary Note 2.

### Confocal microscopy

For staining of fixed cells, 12k cells/well were seeded in tissue culture-treated PhenoPlate 96-well plates (PerkinElmer) and incubated overnight. Cells were transfected with TASL-V5 fragments the next day and subsequently TASL-V5 expression was induced by addition of 1 µg/ml doxycycline and 10 µM C5 or an equivalent volume of DMSO respectively. Compound and doxycycline were added at the same time, to allow C5 to occupy SLC15A4 binding sites in advance of TASL binding. Cells were fixed after 24 h of incubation for 10 min with 4% formaldehyde in PBS and permeabilized and blocked with 0.2% Triton X-100 (Sigma) and 5% FBS in PBS for 1 h. Cells were stained overnight at 4 °C with a mouse anti-V5 primary antibody (Invitrogen) in blocking solution. Cells were washed 3 time in PBS and stained for 1.5 h with fluorescently labeled goat anti-mouse Alexa-568 (Invitrogen) secondary antibody. After 2 more washes, nuclear counterstaining was performed for 10 min with DAPI (Thermo Fisher Scientific), diluted 1:1000 in PBS.

For live cell imaging cells were plated in PhenoPlate 96-well plates or Ibidi m-Slide 8 (Ibidi) wells at 16k/well and incubated over-night. Then compounds were added, if indicated, and cells were imaged after 24 h of incubation. Cells were stained with LysoTracker Deep Red L12492 (Thermo Fisher) and Hoechst 34580 (Thermo Fisher). PhenoPlate images were acquired on an Opera Phenix Plus High-Content Screening System (PerkinElmer) and m-Slide images were acquired on a confocal laser scanning microscope (Zeiss LSM-980, Carl Zeiss) equipped with an incubation chamber. Images were analyzed with Fiji – ImageJ.

### High-throughput screening

TGC cells were plated at 2 k/well on 384-well PerkinElmer Operetta plates preprinted with 10 µM compound in DMSO and incubated for 48 h. Plates were imaged with an Operetta CLS High-Content Analysis system at one field per well and 20x magnification. A pipeline to analyze GFP intensity and mCherry/GFP ratio was set up with Operetta's harmony high-content imaging and analysis software.

### Microscopy-based IRF5- and NF-κB-specific nuclear translocation assay for CAL-1

$5 \times 10^5$ CAL-1 cells in 500 µl medium (wildtype, drug-treated or gene-deficient as indicated) were seeded on poly-L-lysine hydrobromide (P6282, Sigma-Aldrich)-coated coverslips in 24-well cell culture plates and immediately stimulated with 5 µg/ml R848 (Invivogen) or left unstimulated. Afterwards, medium was removed and cells were fixed with 4% formaldehyde in PBS for 15 min at room temperature (RT), followed by blocking and permeabilization with 5% FBS and 0.3% Triton X-100 in PBS for 1 h at RT. Cells were stained with directly conjugated anti-NF-κB p65-Alexa488 (Cell Signaling, no. 49445, 1:200) and anti-IRF5-Alexa647 (Cell Signaling, no. 74818, 1:200) in 1% BSA and 0.3% Triton X-100 in PBS overnight at 4 °C. Slides were rinsed twice with PBS, and nuclear counterstaining was performed with DAPI (Thermo Fisher Scientific, D1306), diluted 1:2,000 in PBS. Stained slides were mounted using ProLong Gold (Thermo Fisher Scientific) antifade reagent.

Images were acquired on a Zeiss LSM-980 equipped with an AiryScan2 detector and processed in Zeiss ZEN version 3.6. Computational image analysis was performed with CellProfiler version 4.2.4 and data analyzed in R version 4.2.1 facilitating the following packages: 'tidyverse', 'data.table' and 'ggpubr'. In brief nuclei and cytoplasmic regions were identified and nuclear to cytoplasmic median intensity ratios were calculated on a single cell level. In the box plots presenting IRF5 or NF-κB nuclear translocation, bars indicate the median, boxes indicate the first to third quartiles. The top whisker extends from hinge to largest value no further than 1.5 × interquartile range (IQR) from the hinge, and the bottom whisker extends from the hinge to smallest value at most 1.5 × IQR of the hinge. Dots indicate outlying points.

### Microscopy-based IRF5- and NF-κB-specific nuclear translocation assay for PBMCs

PBMCs were treated for 24 h with DMSO or C5 (10 µM) and thereafter stimulated with 5 µg/ml R848 (Invivogen) for three hours or left unstimulated. Cells were harvested and stained for 30 min with PE-conjugated anti-CD19 antibodies (BioLegend, #302254, 1:200 in PBS/2% FBS) on ice and fixed with 4% formaldehyde in PBS for 15 min at RT. Permeabilization and staining for NF-κB p65 and IRF5 and counterstaining was performed as for CAL-1 and cells were seeded at a concentration of $3 \times 10^4$ cells/well onto 384-well imaging plates. Images were acquired on a PerkinElmer Opera Phenix HCS spinning disc confocal microscope and analyzed as described above with the additional identification of B cells based on CD19 membrane staining intensity.

## Alphafold modeling of SLC15A4 in complex with TASL

Alphafold predictions of the SLC15A4-TASL complex were generated with Alphafold-multimer[27] and the models were further evaluated visually in PyMol (Schroedinger).

## Cryo-EM structure of SLC15A4-C5 complex

Expression and purification of the SLC15A4/ALFA_Nanobody/C5 complex was performed as for the SLC15A4/ALFA_Nanobody complex described in the accompanying paper by Chen, Xie, Zhang et al., except that additional C5 was used during purification. In brief, 10 µM (final concentration) C5 was added to the lysis buffer, wash buffer, elution buffer and SEC buffer, and 100 µM (final concentration) C5 were added to the peak SEC fractions. Then the fractions were concentrated to 15 mg/ml for grid preparation. The cryo-EM grids for SLC15A4/ALFA_Nanobody/C5 complex were prepared and acquired on a Titan Krios microscope (FEI) as described in Chen, Xie, Zhang et al. For the data set of SLC15A4/ALFA_Nanobody/C5 complex, 4845 micrographs (movie stacks) were collected and processed as in Chen, Xie, Zhang et al. and a 2.50 Å resolution map for the SLC15A4/ALFA_Nanobody/C5 complex was obtained. SLC15A4/ALFA_Nanobody/C5 complex model building was performed using a similar procedure than in Chen, Xie, Zhang et al. Cryo-EM data collection and refinement statistics are shown in Table S2.

## Statistics & reproducibility

Data are represented as individual values, mean, mean ± SD or box plots as described in the figure legends. Group sizes (*n*) are indicated in the figure or figure legends, the applied statistical test in the figure legend. No statistical methods were used to predetermine sample size. No data were excluded from the analyses. The experiments were not randomized and investigators were not blinded to allocation during experiments and outcome assessment.

## Reporting summary

Further information on research design is available in the Nature Portfolio Reporting Summary linked to this article.

# Data availability

The 3D cryo-electron microscopy density map and the coordinates of atomic models has been deposited in the Electron Microscopy Data Bank (EMDB) and the Protein Data Bank (PDB) with the following accession codes: EMDB-36754 and PDB 8JZX for SLC15A4/C5 complex. The coordinates for previously published SLC15A1 used in Figure S10d were obtained from the PDB database accession code 7PMW. All other data are available in the manuscript or in the supplementary materials. Noncommercial reagents described in this manuscript are available from MR, LH or GSF under a material transfer agreement. Source data of relevant information are provided as source data files. Source data are provided with this paper.

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

## Acknowledgements

We would like to acknowledge Georg E. Winter, Christian Löw, and the members of the Superti-Furga, Rebsamen, and Heinz groups for critical discussions and suggestions. We thank Prof. T. Maeda (Nagasaki University) for kindly providing the CAL-1 cells, Johannes W. Bigenzahn for critical reagents, Brigitte Meyer, Carl-Walter Steiner, Birgit Niederreiter, Daniela Sieghart, Irena Stanic and Rebecca Kirkely for technical support. We would like to thank the molecular discovery platform at CeMM for assistance with the small-molecule screen and the imaging core facility of the Medical University of Vienna for assistance with high-resolution imaging. Plasmids obtained through Addgene were a gift from D. Trono, D. Root, and B. Ebert. This work was supported by the Austrian Academy of Sciences (to G.S.-F., A.B., A.K.H., V.D., A.K., R.C., S.K), the Medical University of Vienna (to L.X.H.), the European Alliance of Associations for Rheumatology (EULAR grant RMG2235 to L.X.H. - the content is solely the responsibility of the authors and does not necessarily represent the official views of EULAR), the Austrian Science Fund (Erwin Schrödinger Fellowship FWF3872 to A.B.), the Swiss National Science Foundation (Project grant 310030_200709 to M.R, and Postdoc Mobility Fellowship P500PB_203043 to A.K.H.), the National Key R&D Program of China (2022YFA1302701 to M.Y) and the National Natural Science Foundation of China (32030056 to M.Y).

## Author contributions

A.B., L.H., M.R. and G.S.-F. designed research; A.B., L.B., F.K., H.Z., M.D., A.K., A.H., V.D., M.R. and L.H. performed research, X.C., M.X., S.Z., M.Y. designed, performed and analyzed the cryo-EM research, A.B., L.B., H.Z., L.H., M.R. generated reagents, A.B., A.K., S.K., L.H., M.R. and G.S.-F. analyzed and interpreted the data; A.B., L.B., L.H., M.R. and G.S.-F. wrote the paper with contributions from all other co-authors.

## Competing interests

CeMM and the Medical University of Vienna are the applicants of European priority patent applications (EP 22 203 423.3, EP 22 203 422.5, EP 22 203 421.7, status: filed), with inventors A.B., M.R., L.H. and G.S.-F, covering small-molecule modulators of TASL and their medical use. G.S.-F. and S.K. are co-founders and own shares of Proxygen GmbH and Solgate GmbH, an SLC-focused company. D.A. received grants, speaker fees, or consultancy fees from Abbvie, Gilead, Galapagos, Eli Lilly, Janssen, Merck, Novartis. The other authors declare no competing interests.
