## [Peer Review File · Nature Communications]

Reviewers' Comments:

Reviewer #1:

Remarks to the Author:

This manuscript by Boeszoermyeni et al reports the design of a phenotypic assay for monitoring the interaction between the lysosomal solution carrier protein SLC15A4 and the innate immunity adaptor protein TASL, which is critical for endosomal Toll-like receptor signaling. Using this assay for high-throughput screening, the authors identified a compound named feeblin that inhibits the IRF5-mediated signaling downstream of TLR7/8 through disrupting the SLC15A4-TASL interaction, but does not affect other pathways downstream of TLR. A cryo-EM structure of SLC15A4 bound to feeblin shows that feeblin binds a pocket in the luminal side pocket of SLC15A4 adopting an outward-open conformation, which is incompatible with the binding of TASL to the cytosolic side of SLC15A4. This study therefore establishes a proof-of-principle route for developing allosteric inhibitors of SLC15A4 for autoimmune diseases mediated by the TLR-SLC15A4-TASL-IRF5 pathway. The results presented by the manuscript are thorough and exciting. The manuscript is well written. I only some minor comments:

1. Line 136-137, it is stated that five compounds inhibited IRF5 but not NF- κ B. It would be better to specify which five these are, either in the text or Figure 2a.
2. Figure 3g and h. It appears that the effects of C5 on the expression level and localization of TASL were more pronounced when the co-expressed SLC15A4 was N-terminal truncated and localized to the plasma membrane. Does this suggest that C5 could more easily access the binding pocket in SLC15A4 when it is on the plasma membrane than in lysosome? Or lysosome may actively exclude or degrade C5 to reduce its effect?
3. Is it possible that some cellular endogenous peptides or small molecules bind the C5-binding pocket in SLC15A4 and regulate the SLC15A4-TASL interaction, considering that the residues in the binding pocket are conserved among different species? Related to this point, does C5 bind other SLC15A4 homologues?
4. Line 213, the sentence refers to Fig. 3h, I think this should be Fig. 3i.
5. The FSC curves in the PDB validation reports do not make sense, as the blue unmasked-calculated FSC remaining 1 throughout the spatial frequency ranges. Did the authors upload the two independent half maps?

Reviewer #2:

Remarks to the Author:

Overall, this is an interesting study that explores the druggability of the SLC15A4/TASL interface using a phenotypic drug screening approach. Doing so, the authors identify the inhibitory compound C5 from a large compound library (86,727 small molecules) that acts as a ligand for SLC15A4. Ligand-bound SLC15A4 adopts a conformation that precludes TASL binding from the cytosolic side. This prevents signalling and also shortens the half-life of TASL, thereby reducing its cellular concentration. As a result, endosomal TLRs that signal for IRF5 activation via this interface are blunted in their pro-inflammatory/antiviral response. Taken together, this principle provides a new and intriguing mode of action to subvert the activity of nucleic acid-sensing TLRs, which may be beneficial in certain autoimmune diseases such as SLE.

The experiments are performed to a high standard and the evidence provided clearly supports the conclusions drawn by the authors. For example, the mode of action of C5 binding SLC15A4 is well documented using a cryo-EM structure. I see no major flaws in this paper, just a few minor points that require additional attention.

This work stops at elucidating the mode of action of the lead compound C5 and does not consider additional preclinical studies aimed at developing a new therapeutic drug. This is perfectly acceptable for the identification of a new targeting principle such as the one described here.

Nevertheless, this work could establish C5 as an interesting tool compound for the scientific community to block the activation of TASL-dependent TLRs. Here, some further validation studies would be important to establish the PRR-specificity of this compound. At this stage it appears that C5 has an IC50 of around 1 μ M against TLR7/8 mediated ISRE activation (TASL-dependent), whereas TLR2 dependent activation (TASL-independent) appears to be inhibited at an IC50 of 10 μ M (this is not yet formally established). This would constitute a moderate potency ratio for this comparison, but it would be desirable to assess more pathways.

Therefore, it would be great if the authors could determine the IC50 values of C5 for TLR2 activation (experiments already performed) and also investigate additional pathways such as TLR4, RIG-I, cGAS-STING and/or NOD2 signalling. The more information the authors can provide on the specificity of their compound, the more useful it would be for the community. In this context, it would also be great if the authors could investigate a cytokine readout rather than a reporter assay and also use primary cells rather than cell lines.

Reviewer #3:

Remarks to the Author:

This article is a follow-up extension of the authors' work published in Nature in 2020. Based on the previous findings that SLC15A4-TASL interactions are involved in regulating the TLR7/8-IRF5 pathway associated with systemic lupus erythematosus (SLE), the authors targeted the TASL protein stability screen affected by SLC15A4-TASL interactions to obtain an IRF5 pathway-specific inhibitor feebelin, and demonstrated in normal human and SLE patient cells that feebelin selectively inhibits the IRF5 pathway via SLC15A4, and revealed by resolving the structure of C5 and SLC15A4 that feebelin locks SLC15A4 in a conformation that is unable to bind TASL. The study represents a proof-of-concept for the development of new therapeutics for SLE and related diseases by targeting SLC15A4-TASL .

Some advice and questiones:

1. Format: The abstract — should be no more than 150 words long and contain no references.
2. Line 200:what does "PM" represent?
3. Line 213:Figure citation mistake. It seems to be Fig.3i.
4. Figure 2a: The rationale for picking these five compounds over others is puzzling, as C3 and C6, for example, also seem to meet the selection criteria.
5. Some grammar mistakes: line 63 'the highest-ranked models' and line 126 'secondary assays'.

The readers might be interested in this study. I recommend accepting the article with minor revision.

Reply to Reviewers - NCOMMS-23-12891-T: Boeszoermyeni et al. "A conformation-locking inhibitor of SLC15A4 with TASL proteostatic anti-inflammatory activity"

We thank the Reviewers for the careful and constructive assessment of our study. We are grateful for the insightful comments, which we have addressed in the detailed point-by-point reply here below and in the revised manuscript. We believe that the suggested experiments strongly contributed to further improving our study.

REVIEWER COMMENTS

Reviewer #1 (Remarks to the Author):

This manuscript by Boeszoermyeni et al reports the design of a phenotypic assay for monitoring the interaction between the lysosomal solution carrier protein SLC15A4 and the innate immunity adaptor protein TASL, which is critical for endosomal Toll-like receptor signaling. Using this assay for high-throughput screening, the authors identified a compound named feeblin that inhibits the IRF5-mediated signaling downstream of TLR7/8 through disrupting the SLC15A4-TASL interaction, but does not affect other pathways downstream of TLR. A cryo-EM structure of SLC15A4 bound to feeblin shows that feeblin binds a pocket in the luminal side pocket of SLC15A4 adopting an outward-open conformation, which is incompatible with the binding of TASL to the cytosolic side of SLC15A4. This study therefore establishes a proof-of-principle route for developing allosteric inhibitors of SLC15A4 for autoimmune diseases mediated by the TLR-SLC15A4-TASL-IRF5 pathway. The results presented by the manuscript are thorough and exciting. The manuscript is well written. I only some minor comments:

We thank the Reviewer for the positive assessment of our study and the constructive suggestions to further improve it, which we have fully addressed in the revised manuscript as detailed here below.

1. Line 136-137, it is stated that five compounds inhibited IRF5 but not NF-kB. It would be better to specify which five these are, either in the text or Figure 2a.

We have now specified the five compounds (C2, C4, C5, C8 and C10) in the revised manuscript (page 5, line 148).

2. Figure 3g and h. It appears that the effects of C5 on the expression level and localization of TASL were more pronounced when the co-expressed SLC15A4 was N-terminal truncated and localized to the plasma membrane. Does this suggest that C5 could more easily access the binding pocket in SLC15A4 when it is on the plasma membrane than in lysosome? Or lysosome may actively exclude or degrade C5 to reduce its effect?

We thank the Reviewer for this insightful comment and, indeed, we agree with the observation that C5 seems to be more active towards the plasma membrane-localized SLC15A4. Given that C5 binds SLC15A4 in the outward open conformation, it is likely that both the availability of the compound and the accessibility of the binding pocket are the highest at the PM, while it is possible that the uptake and/or delivery of C5 to the endolysosomal lumen could result in a local concentration of C5 that is lower than in the extracellular milieu.

Although we favor the above-mentioned hypothesis, we cannot exclude at this point that, as suggested by the Reviewer, other mechanisms such as drug efflux or lysosomal metabolization of C5 could also contribute to the phenomenon.

We have now elaborated on these interesting hypotheses in the revised discussion (page 10, lines 298-303)

3. Is it possible that some cellular endogenous peptides or small molecules bind the C5-binding pocket in SLC15A4 and regulate the SLC15A4-TASL interaction, considering that the residues in the binding pocket are conserved among different species? Related to this point, does C5 bind other SLC15A4 homologues?

We thank the Reviewer for pointing out this highly interesting and relevant point. We are also convinced that the exquisiteness of the allosteric regulation exploited by C5 most likely reflects a physiological mechanism. As mentioned by the Reviewer, one can imagine a lysosomal metabolite that acts by modulating the TLR7/9 signaling output, perhaps in a pH-dependent manner.

Interestingly, SLC15A4 is broadly expressed, including in cell types that do not co-express TASL nor endosomal TLRs. As previously reported, it is therefore likely that SLC15A4 harbors transport activity towards dedicated substrates such as histidine and/or oligopeptides, requiring a conserved binding pocket. In terms of endolysosomal TLR responses, the transport activity is dispensable once TASL is artificially localized to lysosomes (see Fig. 3i, j and ref. 30 - Zhang et al, Cell Reports 2023). This does not exclude, as mentioned by the referee, that SLC15A4 transport activity and/or binding of an endogenous metabolite could modulate SLC15A4 conformation and thereby TASL recruitment. Indeed, any condition stabilizing the outward or inward conformation is expected to inhibit or promote TASL recruitment, respectively. The discovery of such an endogenous ligand will be an important breakthrough that is likely to showcase the exciting intersection of metabolism and immunity. The identification is likely to use experimental systems that preserve physiology as much as possible and could exploit C5 or its fluorescent derivatives in the discovery strategy.

Concerning the question of binding of C5 to SLC15A4 homologs, our sequence analysis suggests that the binding pocket is conserved between SLC15A4 and SLC15A3, therefore potential engagement of C5 by SLC15A3 could not be excluded. Of note, despite the high degree of similarity, our previous data clearly show that SLC15A4, but not SLC15A3, is competent for TASL binding and that SLC15A3 is not able to functionally complement SLC15A4 deficiency (ref. 23 - Heinz et al, Nature 2020). Moreover, clear evidence indicates that SLC15A3 is dispensable for endosomal TLR9 responses in primary murine pDCs (ref. 32 - Rimann et al, PNAS 2022) Importantly, our data (Fig. 3i, j), showing that C5 does not impair IRF5 activation and cytokine production in SLC15A4-deficient cells expressing TASL artificially localized to lysosomes, strongly support that any potential effect on other solute carriers is not relevant for the TLR-inhibiting effect of C5/feeblin. As the compound came directly out of a screened library and no optimization has occurred yet, we are confident that it will be possible to address any possible issues concerning selectivity, potency as well as many other parameters in the future.

We have now included these important considerations in the revised discussion (pages 10 and 11).

4. Line 213, the sentence refers to Fig. 3h, I think this should be Fig. 3i.

We apologize for this oversight and thank the Reviewer for identifying the error, which we corrected in the revised manuscript.

5. The FSC curves in the PDB validation reports do not make sense, as the blue unmasked-calculated FSC remaining 1 throughout the spatial frequency ranges. Did the authors upload the two independent half maps?

We would like to thank the reviewer for pointing out this mistake in the validation report and have now obtained a new validation report, in which the FSC curve converges correctly. Since we had to re-submit the PDB and data files we obtained a new code for the SLC15A4/C5 complex structure: PDB

8JZX and EMDB 36754. Please find enclosed with this submission the revised PDB validation report for the SLC15A4/C5 structure. We also include now a new supplementary figure presenting the reconstruction and structure determination of human SLC15A4/C5 complex (New Fig. S8).

Reviewer #2 (Remarks to the Author):

Overall, this is an interesting study that explores the druggability of the SLC15A4/TASL interface using a phenotypic drug screening approach. Doing so, the authors identify the inhibitory compound C5 from a large compound library (86,727 small molecules) that acts as a ligand for SLC15A4. Ligand-bound SLC15A4 adopts a conformation that precludes TASL binding from the cytosolic side. This prevents signalling and also shortens the half-life of TASL, thereby reducing its cellular concentration. As a result, endosomal TLRs that signal for IRF5 activation via this interface are blunted in their pro-inflammatory/antiviral response. Taken together, this principle provides a new and intriguing mode of action to subvert the activity of nucleic acid-sensing TLRs, which may be beneficial in certain autoimmune diseases such as SLE.

The experiments are performed to a high standard and the evidence provided clearly supports the conclusions drawn by the authors. For example, the mode of action of C5 binding SLC15A4 is well documented using a cryo-EM structure. I see no major flaws in this paper, just a few minor points that require additional attention.

We are grateful for the positive assessment of our work, and we thank the Reviewer for the constructive suggestions, which we have addressed in the revised manuscript as detailed here below.

This work stops at elucidating the mode of action of the lead compound C5 and does not consider additional preclinical studies aimed at developing a new therapeutic drug. This is perfectly acceptable for the identification of a new targeting principle such as the one described here. Nevertheless, this work could establish C5 as an interesting tool compound for the scientific community to block the activation of TASL-dependent TLRs. Here, some further validation studies would be important to establish the PRR-specificity of this compound. At this stage it appears that C5 has an IC₅₀ of around 1 μM against TLR7/8 mediated ISRE activation (TASL-dependent), whereas TLR2 dependent activation (TASL-independent) appears to be inhibited at an IC₅₀ of 10 μM (this is not yet formally established). This would constitute a moderate potency ratio for this comparison, but it would be desirable to assess more pathways.

Therefore, it would be great if the authors could determine the IC₅₀ values of C5 for TLR2 activation (experiments already performed) and also investigate additional pathways such as TLR4, RIG-I, cGAS-STING and/or NOD2 signalling. The more information the authors can provide on the specificity of their compound, the more useful it would be for the community. In this context, it would also be great if the authors could investigate a cytokine readout rather than a reporter assay and also use primary cells rather than cell lines.

We thank the Reviewer for appropriately assessing the scope our study. As mentioned in the original discussion and acknowledged by the Reviewer, our work aimed at demonstrating the “druggability” of the SLC15A4-TASL complex. The identification of C5 provided this proof-of-principle and revealed a novel targeting mode, supported by functional and structural data. Indeed, C5 is the original compound identified directly out of our screening library without any further optimization. As such we think it is remarkable that it shows activity already at a single digit micromolar concentration. It is also remarkable that its kinetic parameters allowed to determine its complex with SLC15A4 by cryoEM. We therefore consider C5 as an excellent starting point for future hit/lead optimization efforts aiming at

increasing its potency, selectivity and specificity. These perspectives for future developments are now highlighted in the updated discussion (pages 10 and 11).

*As suggested by the Reviewer we have now also further explored its specificity for TLR7-9 versus other PRRs. Concerning TLR2 responses, we have now added the IC50 values in Fig 2b, which indicated 10-30 μ M, as correctly estimated by the Reviewer. We next assessed the effect of C5 in THP1 DUAL cells using a broader panel of innate immune stimuli activating ISRE- and/or NF- κ B-driven responses (New Fig. S3c). In line with data obtained for R848 (TLR7/8), C5 similarly inhibited ISRE-driven responses induced by the weaker TLR8 agonist ssRNA40. Furthermore, we confirmed reduced potency on Pam3CSK4 (TLR2)-induced responses and showed even lower activity upon stimulation with cGAMP (STING agonist) or transfected poly(I:C) (RLR ligand) (New Fig. S3c). Moreover, C5 had virtually no effect on NF- κ B reporter activity induced by C12-*ie*DAP or L18-MDP (lipidated agonists of NOD1 and NOD2 respectively) (New Fig. S3c). Lastly, we assessed the effect of C5 on diverse PRR pathways using cytokine/chemokine production (TNF and CCL5) as read-out (New Fig. 2i and Fig. S4g). In line with the reporter assay, C5 impaired R848, had modest effect on Pam3CSK4 responses and manifested no inhibition upon stimulation by cGAMP or LPS (New Fig. 2i and Fig. S4g).*

Altogether, these data indicate that C5 exerts strong specificity for endolysosomal TLR7-9 responses, compared to other PRR-pathways. As discussed above, we are confident that this could be further improved by structure-function optimization of the compound to increase potency, selectivity and specificity to reduce the effects on other PRR responses and possible lysosomal effects.

Reviewer #3 (Remarks to the Author):

This article is a follow-up extension of the authors' work published in Nature in 2020. Based on the previous findings that SLC15A4-TASL interactions are involved in regulating the TLR7/8-IRF5 pathway associated with systemic lupus erythematosus (SLE), the authors targeted the TASL protein stability screen affected by SLC15A4-TASL interactions to obtain an IRF5 pathway-specific inhibitor feeblin, and demonstrated in normal human and SLE patient cells that feeblin selectively inhibits the IRF5 pathway via SLC15A4, and revealed by resolving the structure of C5 and SLC15A4 that feeblin locks SLC15A4 in a conformation that is unable to bind TASL. The study represents a proof-of-concept for the development of new therapeutics for SLE and related diseases by targeting SLC15A4-TASL.

The readers might be interested in this study. I recommend accepting the article with minor revision.

We thank the Reviewer for the positive assessment of our study. We have now included her/his suggestions in the revised manuscript as detailed here below.

Some advice and questions:

1. Format: The abstract — should be no more than 150 words long and contain no references.

We have now reformatted the abstract accordingly to Nature Communications guidelines.

2. Line 200: what does “PM” represent?

PM stands for “plasma membrane”, we have now included this missing information in the revised manuscript.

3. Line 213: Figure citation mistake. It seems to be Fig.3i.

We apologize for this oversight and thank the reviewer for identify the error, which we corrected in the revised manuscript.

4. Figure 2a: The rationale for picking these five compounds over others is puzzling, as C3 and C6, for example, also seem to meet the selection criteria.

After evaluation of the initial 12 compounds in our IRF5/NF- κ B nuclear translocation assay, we reduced the number to 5 in order to allow for a more detailed characterization. For this, we selected those compounds with the strongest effect on R848-induced IRF5 nuclear translocation, integrating also visual inspection of the cellular status and preliminary THP1 dual assays. Concerning specifically C3 and C6, their activity was below the cut-off of the top 5 candidates (Fig 2a). Moreover, preliminary investigations suggested lower potency in THP1 dual assay compared to the selected compounds and some level of toxicity (data not shown), while for C6 the reduction of endogenous TASL protein levels also appeared to be minimal (Figure S2a).

5. Some grammar mistakes: line 63 'the highest-ranked models' and line 126 'secondary assays'.

We apologize for these mistakes, which have been corrected in the revised manuscript.

Reviewers' Comments:

Reviewer #1:

Remarks to the Author:

The authors have addressed my concerns.

Reviewer #2:

Remarks to the Author:

My questions have been answered sufficiently. I have no further comments.

Reply to Reviewers - NCOMMS-23-12891-T: Boeszoermyeni et al. *“A conformation-locking inhibitor of SLC15A4 with TASL proteostatic anti-inflammatory activity”*

We thank the Reviewers for the careful and constructive assessment of our study and are pleased that we could address all concerns sufficiently.

REVIEWERS' COMMENTS

Reviewer #1 (Remarks to the Author):

The authors have addressed my concerns.

Reviewer #2 (Remarks to the Author):

My questions have been answered sufficiently. I have no further comments.